# Stability of the Communication Function Classification System among Children with Cerebral Palsy in South Korea

**DOI:** 10.3390/ijerph18041881

**Published:** 2021-02-15

**Authors:** Eun-Young Park

**Affiliations:** Department of Secondary Special Education, College of Education, Jeonju University, Jeonju 55069, Korea; eunyoung@jj.ac.kr; Tel.: +82-63-220-3186

**Keywords:** children with cerebral palsy, agreement, stability, communication function classification system

## Abstract

Interest in the prognosis of skill levels has been an important issue among children with cerebral palsy (CP). This study aimed to verify the stability of the Communication Function Classification System (CFCS) in 2- to 18-year-old children with CP. Data collected from 171 children with CP who received rehabilitation therapy in hospitals or attended special elementary schools in South Korea were reviewed. They were divided into two groups, children <4 years and children ≥4 years. Participants were evaluated over 1-year and 2-year intervals from the first rating. Agreement between the three measurements and the weighted kappa were analyzed. At the 1-year interval, results demonstrated a high agreement rate of the CFCS in children ≥4 years old, and during the 2-year interval the study revealed a low agreement rate in children aged 2–4 years. The results indicated the stability of the CFCS in children ≥4 years old but some change of the CFCS in 2- to 4-year-old children. Moreover, the findings suggested that the change of the CFCS varied with time and age. Based on these results, it is recommended that the CFCS assessments be performed periodically, especially among 2- to 4-year-old children with CP.

## 1. Introduction

For children with cerebral palsy (CP), it is essential to assess their communication abilities to better support their needs. However, evaluating the communication function of children with CP is difficult to do accurately with a single evaluation system. In the case of children with CP, evaluating items included in communication measures poses many challenges, including practicality and familiarity with the content of assessment and/or evaluation measures. Therefore, there is a need for a comprehensive assessment that uses an informal evaluation based on direct observation or parent and teacher feedback [1] and comprehensive tools in a natural setting.

Communication among children with CP is one of relevant activities identified by the International Classification of Functioning, Disability, and Health’s (ICF) conceptual frameworks. Communication activities include the transmission and reception of messages such as speaking, listening, reading, writing, and the use of alternative and augmentative communications [2,3]. The incidence of communication disorders in children with CP has been reported to vary widely. The Communication Function Classification System (CFCS) [4] that is used to classify the communication level in individuals with CP was developed based on the Gross Motor Function Classification System (GMFCS) [5] and the Manual Ability Classification System (MACS) [6] as part of a growing trend in classifying the activities and impairments proposed by the ICF [4]. Since they were developed, the psychometric properties of the classification systems have been actively studied. The reliability and validity of the GMFCS and MACS have been reported, and studies on the validity of the CFCS [4] have also been reported. Sixty-one experts evaluated 68 children with CP and aged 2 to 18 years on the CFCS and reported a reliability of 0.66 (95% CI = 0.55–0.78), a test-retest reliability of 0.82 (CI = 0.74–0.90), and an inter-rater reliability of 0.49 (95% CI = 0.40–0.59) [4]. 

The Korean translation of the CFCS [7] was used to evaluate its reliability. The results showed that the test-retest reliability among professionals was 0.991 (95% CI = 0.979–1.00), the inter-rater reliability among professionals was 0.905 (95% CI = 0.864–0.946), and the inter-rater reliability between the professional and parent groups was 0.882 (95% CI = 0.837–0.927) [7]. One of the important psychometric properties of the classification systems is stability over time. In the case of the GMFCS, 610 children with CP were confirmed to be stable [8]. The MACS also confirmed the stability of 1267 children with CP over five years and confirmed the stability of the MACS level over time [9]. In the case of the first GMFCS, research has been conducted on whether the classification system is stable over time [10,11], and subsequent studies on the MACS provide information on the change [9]. However, the relatively recent developments in the CFCS have not been tested for stability or change as children grow. Recently, one study [12] examined the stability of the CFCS with GMFCS and MACS in 664 children with CP (with ages ranging from 18 months to 12 years). It was reported that the kappa coefficients varied from 0.76–0.88 for the GMFCS, 0.59–0.73 for the MACS, and 0.57–0.77 for the CFCS. Whether or not the child’s skill level will change because of a change in prognosis, decision making and counseling with parents is an important issue, and the degree of stability of the classification system can provide information regarding the possibility of changes in functioning in the child. The stability of the classification system indicates whether children with CP maintain the same level of functioning over time or whether they can be reclassified to different levels over time [13]. Considering the lack of related information on the stability of the CFCS, this study aimed to determine its stability over a second rating and third rating from the first rating.

## 2. Materials and Methods

### 2.1. Participants

Totally, 171 children with CP (mean = 10.9 years, SD = 4.6 years) participated in this study. Participants attended a convalescent or rehabilitation center for disabled individuals or a special school for physical disabilities in South Korea. There were 99 boys (57.9%) and 72 girls (42.1%). The age range was 2 to 18 years. Totally, 21 children with CP were below 4 years and 150 were older. The parents of all children agreed to participate in this study. The types of CP in the children were spastic (81.0%), dyskinetic/athetotic (6.8%), ataxic (3.4%), and hypotonic (8.8%). The participants were classified using the GMFCS: 19 (11.1%) were classified into Level I, 24 (14.0%) into Level II, 18 (10.5%) into Level III, 24 (14.0%) into Level IV, and 86 (50.3%) into Level V. The present study was approved by the Research Ethics Board of Jeonju University (Jeonju University IRB-1041042-2013-1).

### 2.2. Measurement

To classify their communication function, the CFCS was used. CFCS is a tool developed with support from the National Institute of Health (NIH), the Cerebral Palsy International Research Foundation, and The Hearst Foundation. We used the Korean version of the CFCS (http://cfcs.us/wp-content/uploads/2018/11/CFCS_Korean.pdf) [7] developed by Hidecker and her colleagues [4]. The CFCS content was as follows: Level 1 child with CP is an effective sender and receiver with unfamiliar and familiar partners. Level 2 child with CP is an effective but slower paced sender and/or receiver with unfamiliar and familiar partners. Level 3 child with CP is an effective sender and receiver with familiar partners. Level 4 child with CP is sometimes an effective sender and receiver with familiar partners. Level 5 child with CP is seldom an effective sender and receiver even with familiar partners. The inter-rater reliability of CFCS has been reported from 0.66–0.98 [4,14]. Hidecker et al. [15] reported the validity of the CFCS in preschool children with speech and language disorders. The stability of the CFCS was reported by previous studies. Palisano et al. [12] reported the kappa coefficients to be 0.57–0.77 in 664 children with CP.

### 2.3. Raters

Data regarding the CFCS were collected by 18 occupational therapists who had treated children with CP for more than 6 months. One occupational therapist evaluated 3–9% of the children with CP. Although occupational therapists met the eligibility requirements for administering the CFCS evaluations, all therapists were trained specifically for using these evaluation tools. They were encouraged to seek clarification from the investigators regarding any question or problem arising during the assessments. The amount of time it took to complete the CFCS was less than 5 min because the occupational therapists were familiar with children with CP. The same evaluators performed the assessments across the three measurements and were blinded to the previous CFCS ratings.

### 2.4. Data Collection Procedure

The data were collected once a year over two years during the same month. For example, if data were collected on 1 September of the first year, data were collected between 1 September and 31 September the next year. After the third measurement of evaluation, all collected data were subjected to central statistical monitoring.

### 2.5. Statistical Method

Agreement rate of the CFCS and weighted kappa were used in this study to examine the stability of the CFCS in three repetitive ratings in two years. Weighted kappa was performed to investigate the change of the CFCS across three times measurements. In general, the criteria for the weighted kappa are: a slight agreement between 0 and 0.20, a fair agreement between 0.21 and 0.40, a moderate agreement between 0.41 and 0.60, a substantial agreement between 0.61 and 0.80, and 0.81 to 1.00 interprets as an almost perfect agreement [16]. Agreement rate of stability above 80% was of acceptable value [17].

Based on a previous study on CFCS’s stability, here the CFCS stability was verified by evaluating data for children ≥4 years versus children <4 years [12]. The analysis was completed for about 171 children (whole group) with 21 being below 4 years (younger group) and 150 above 4 years (older group). The age criterion for dividing two subgroups was set to 4 years with reference to the first study to find out the stability of the CFCS [12]. The age reference point for dividing the groups was 4 years at first rating.

## 3. Results

### 3.1. Agreement Rate across Three Measurement Points

The agreement rate between the first and second ratings according to age ranges is shown in Table 1. In the whole group, the agreement rate between the first and second ratings was 69.9% (*n* = 118). In total, 30 children with CP (17.6%) were rated in the more severe functional level and 23 (13.5%) were rated in the less severe functional level during the second rating than in the first rating. Regarding the younger group, the change rate was 33.3% (*n* = 7). Three children with CP (14.3%) were rated in the more severe functional level and four children with CP (19.0%) were rated in the less severe functional level during the second rating than in the first rating. Regarding the older group, the change rate was 30.7% (*n* = 46). Totally, 27 children with CP (18.0%) were rated in the more severe functional level and 19 (12.7%) were rated in the less severe functional level.

The agreement rates between the first and third ratings among different age groups are shown in Table 2. In the whole group, the agreement rate between the first rating and ratings was 66.1% (*n* = 113). In total, 21 children with CP (12.3%) were rated in the more severe functional level and 37 (21.6%) were rated in the less severe functional level during the third rating than in the first rating. Regarding the younger group, the change rate was 42.9% (*n* = 9). Four children with CP (19.0%) were rated in the more severe functional level and five children with CP (23.9%) were rated in the less severe functional level during the third rating than in the first rating. Regarding the older group, the change rate was 28.7% (*n* = 49). Totally, 26 children with CP (17.4%) were rated in the more functional severe level and 23 (15.3%) were rated in the less severe functional level.

The agreement rate between the second and third ratings according to age ranges is shown in Table 3. In the whole group, the agreement rate between the second and third ratings was 80.1% (*n* = 137). In total, 13 children with CP (7.6%) were rated in the more severe functional level and 21 (13.3%) were rated in the less severe functional level during the third rating than in the second rating. Regarding the younger group, the change rate was 33.3% (*n* = 7). Two children with CP (9.5%) were rated in the more severe functional level and five children with CP (23.8%) were rated in the less severe functional level during the third rating than in the second rating. Regarding the older group, the change rate was 18.0% (*n* = 27). Totally, 11 children with CP (7.3%) were rated in the more severe functional level and 26 (10.7%) were rated in the less severe functional level.

### 3.2. Agreement across Three Ratings

In terms of the CFCS levels’ stability, the weighted kappa coefficients were 0.757 to 0.873 in the whole group (Table 1, Table 2 and Table 3). The lowest coefficient was between the first and third ratings measurement points in the younger group. The highest coefficient was between the second and third ratings in the older group. Appearance patterns of the weighted kappa were the same in all age groups.

## 4. Discussion

The functional classification systems in children with CP are widely used not only in research, but also in clinical practice. This aligns with a WHO proposed functioning and disability assessment approach focused on activities and participation restrictions [18]. The CFCS for functional classification of communicative ability in children with CP has been recently developed and has been built on five levels to correspond to the well-known GMFCS for motor performance. Our study was conducted to evaluate the stability of the CFCS, which was developed to classify the level of communication function in children with CP. Possessing information on the functional state of children with CP can help in improving the quality of life of these children and their families, ensuring them a promising future [19].

The weighted kappa coefficients, the primary measure of stability in our study, provide evidence of the stability of the CFCS for 4-year to 18-year-old children with CP in one and two-year intervals and for children below 4 years and with CP between the first and third ratings based on the above 0.75 value. There was no stability of the CFCS for children below 4 years and with CP between the first rating and second rating and the first rating and third rating. According to the criterion, the results of this study showed the almost perfect agreement of the CFCS for children above 4 years and with CP between the second and third ratings. The weighted kappa results showed higher value than in previous studies that examined the stability of the CFCS for children with CP. Palisano et al. [12] reported that the linear weighted kappa was 0.57 for children below 4 years and with CP in a 12 month visit and 0.77 for children above 4 years and with CP.

The weighted kappa for the CFCS was lower than the GMFCS and MACS. Over 0.80 weighted kappa for the GMFCS have been reported by previous studies. The weighted kappa of 0.895 was reported in 103 participants aged 17–38 years [11], and Palisano et al. [8] reported that the weighted kappa coefficient for the GMFCS between the first and last measurements was 0.84 and 0.89 for children <6 years old and at least 6 years old, respectively. Palisano et al. [12] reported that the weighted kappa was 0.76 to 0.95 from the GMFCS and 0.59 to 0.73 from the MACS. The first reason for the lower weighted kappa might be due to the characteristics of the CFCS. The CFCS derives one overall rating based on subjective judgments about how well your child sends messages, how quickly you communicate, and how well your child receives or understands your messages [4]. The need to resolve a single CFCS score to account for these skills is inconsistent between the level of expressiveness and capacity-communication, or there is an expressive repertoire of skills, but it is problematic for children with slow communication due to motorized speech or augmentative and alternative communication (AAC) access [20]. The second reason might be due to the characteristics of communication. Since communication is not a single function, it is not possible to capture the multidimensional nature of communication with a single five-stage measurement, suggesting a breakdown of the communication classification into component functions [21]. The findings of this study on the CFCS suggest a challenge for further studies that need to find out why the kappa is lower than other classification systems and what is the way to increase it.

The agreement rate of this study was 66.9% between the first and third ratings and 80.1% between the second and third ratings. This rate showed a different agreement rate in relation to previous studies. The agreement rate of children below 4 years and with CP between the first and third ratings (57.1%) was higher than in the study by Palisano et al. [12] (51.6%). Similarly, the agreement rate of children above 4 years and with CP between the first and second ratings (67.3%) was higher than in the study by Palisano et al. [12] (64.5%). The agreement rate results were context with the linearly weighted kappa results, which was lower than other classification systems. The original CanChild study of the GMFCS stability had a higher agreement of 76% and 83% for children younger and older than 6 years, respectively [8]. One of the possible reasons that the results of this study are not completely consistent with previous studies could be the environmental impact of communication. The level of CFCS changed to a more severe functional level and a less severe functional level. In this study, the number of children with CP that moved into a 2-level difference was seven both between the first and second rating and also between the first and third rating. The number of children with a 2-level difference between the second and third rating was five. The case that moved into a less severe functional level was higher than those that moved into a more severe functional level. As a factor influencing the change to a less severe functional level, the effect of child maturity and intervention is expected. It is necessary to also consider the development of communication skills through maturity as a cause that may affect the change in the level of CFCS in children. If the child received AAC-based interventions or other effective speech and language interventions, this would have affected the less severe functional changes in CFCS levels. As a factor influencing the change to a more severe functional level, post-seizures, other co-morbidities and losing access to AAC could be a possibility. Hidecker et al. [22] reported that seizures and other co-morbidities have a negative relation with communication function in children with CP. The number of reclassified children, especially those below 4 years, indicates that children with CP do not always maintain the same level of function. Although a 3-year to 5-year period was recommended for children and young people aged 4 to 17 with regards to the GMFCS re-evaluation period, the change rate of the CFCS found between the first and third ratings in this study may suggest a need to re-rate the participants every two years. Especially, short period re-rate of the CFCS for children below 4 years and with CP might be needed for monitoring their communication ability.

Although this study provided information on the stability of the CFCS through observations at intervals of one and two years, from the first rating, there are some limitations. First, the pilot study about the inter-rater reliability or test-retest reliability of the CFCS of this study was not completed and demographic characteristics of the assessors of the CFCS were not described. Second, there was a possibility that the occupational therapists could be misclassifying CFCS levels for children below 4 years. Third, the number of children below 4 years was relatively small and there was a high proportion of children with GMFCS level 5 observed and this proportion was higher than in a previous study [23]. Since children with severe CP showed that the stability of the functional classification might be better, the results of this study should be interpreted in consideration of the severity. Fourth, there were cases of the increase and decrease in GMFCS level, however, the results lack specificity. In future studies, it will be necessary to investigate variables that affect large functional changes.

## 5. Conclusions

Although the results of this study showed overall substantial agreement of the CFCS, regular re-evaluation of the CFCS levels is necessary. The differences in each research periods between the first and third ratings, with one-year intervals, suggested that there were some differences in the stability of the CFCS according to the children’s ages. In future research, systematic and periodic evaluation of the CFCS levels is warranted for verifying the differences in the change rate according to age. In addition to the periodical evaluation, it is considered necessary to include changes in communication methods and comorbid diseases such as seizures after communication intervention. The results showed that the CFCS levels in children with CP increased and decreased in this study, but the factors that influenced these changes should be confirmed in further studies. Understanding why children’s CFCS levels change can help with their prognosis. In addition, information on how the communication function of children with CP develops over time will help the clinician plan treatment strategies accordingly. Finally, insight into the aging-related impact on communication function in children with CP can be used to develop policies and programs that can help prepare such children for adulthood.

## Figures and Tables

**Table 1 ijerph-18-01881-t001:** The CFCS levels between the first rating and second rating.

Category		Second Rating					
		Level I	Level II	Level III	Level IV	Level V	Total
Whole group (Age from 2 to 18)							
First rating	I	20	4	0	0	0	24
	II	3	21	5	2	0	31
	III	3	2	9	9	0	23
	IV	0	2	2	25	10	39
	V	0	0	2	9	43	54
	Total	26	29	18	45	53	171
	Percentage agreement 69.9%, Weighted kappa 0.775 (95% CI 0.717~0.833)						
Younger subgroup (Age below 4)							
First rating	I	1	0	0	0	0	1
	II	0	5	1	1	0	7
	III	0	2	2	0	0	4
	IV	0	0	0	2	1	3
	V	0	0	1	1	4	6
	Total	1	7	4	4	5	21
	Percentage agreement 66.7%, Weighted kappa 0.702 (95% CI 0.492~0.913)						
Older subgroup (Age above 4)							
First rating	I	19	4	0	0	0	23
	II	3	16	4	1	0	24
	III	3	0	7	9	0	19
	IV	0	2	2	23	9	36
	V	0	0	1	8	39	48
	Total	25	22	14	41	48	150
	Percentage agreement 69.3%, Weighted kappa 0.782 (95% CI 0.722~0.842)						

**Table 2 ijerph-18-01881-t002:** The CFCS levels between the first rating and second rating.

Category		Third Rating					
		Level I	Level II	Level III	Level IV	Level V	Total
Whole group (Age from 2 to 18)							
First rating	I	20	4	0	0	0	24
	II	5	17	8	1	0	31
	III	4	2	9	8	0	23
	IV	0	2	1	27	9	39
	V	0	0	2	12	40	54
	Total	29	25	20	48	49	171
	Percentage agreement 66.1%, Weighted kappa 0.757 (95% CI 0.699~0.815)						
Younger subgroup (Age below 4)							
First rating	I	1	0	0	0	0	1
	II	0	5	2	0	0	7
	III	2	0	1	1	0	4
	IV	0	0	0	2	1	3
	V	0	0	1	2	3	6
	Total	3	5	4	5	4	21
	Percentage agreement 57.1%, Weighted kappa 0.620 (95% CI 0.410~0.831)						
Older subgroup (Age above 4)							
First rating	I	19	4	0	0	0	23
	II	5	12	6	1	0	24
	III	2	2	8	7	0	19
	IV	0	2	1	25	8	36
	V	0	0	1	10	37	48
	Total	26	20	16	43	45	150
	Percentage agreement 67.3%, Weighted kappa 0.774 (95% CI 0.714~0.833)						

**Table 3 ijerph-18-01881-t003:** The CFCS levels between the second rating and third rating.

Category		Third Rating					
		Level I	Level II	Level III	Level IV	Level V	Total
Whole group (Age from 2 to 18)							
Second rating	I	24	2	0	0	0	26
	II	5	20	3	1	0	29
	III	0	3	11	4	0	18
	IV	0	0	6	36	3	45
	V	0	0	0	7	46	53
	Total	29	25	20	48	49	171
	Percentage agreement 80.1%, Weighted kappa 0.873 (95% CI 0.832~0.914)						
Younger subgroup (Age below 4)							
Second rating	I	1	0	0	0	0	1
	II	2	4	1	0	0	7
	III	0	1	2	1	0	4
	IV	0	0	1	3	0	4
	V	0	0	0	1	4	5
	Total	3	5	4	5	4	21
	Percentage agreement 66.7%, Weighted kappa 0.774 (95% CI 0.628~0.920)						
Older subgroup (Age above 4)							
Second rating	I	23	2	0	0	0	25
	II	3	16	2	1	0	22
	III	0	2	9	3	0	14
	IV	0	0	5	33	3	41
	V	0	0	0	6	42	48
	Total	26	20	16	43	45	150
	Percentage agreement 82.0%, Weighted kappa 0.885(95% CI 0.843~0.927)						

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
