# Peer review of "Stability of the Communication Function Classification System among Children with Cerebral Palsy in South Korea"

_ijerph, 2021, doi:10.3390/ijerph18041881_

Round 1

Reviewer 1 Report

The article concerns the stability of the Communication Function Classification System (CFCS) in a limited group of South Korea children with cerebral palsy (n=171). It repeats the only study present in literature about this topic made by Palisano et al. (2018) on a larger sample of 664 children. The authors cited Palisano et al. in the article and they  got inspired to their methodology.

The results are similar but not completely overlapped an to my opinion the  reasons of these different results are not well explained. Probably about communication there are also enviromental factors that must be keep in mind 

Moreover, since that the South Korea sample is limited respect to that reported by Palisano et al., I suggested to modify the title specifying that it regards  only that population.

There are many mistakes about English that need a revision of the language (very often authors speak about children and use " was" instead of "were" in particular in the description of the tables).

There is a repetition at line 73 page 2 and a double verb at line 57 page 2 (are have).

The authors must change the way to insert the reference at line 163 page 5 and line 34 page 1.

I missed the sense of the phrase at line 186 page 6

Author Response

In the revised manuscript, I have highlighted in blue where I have made changes; please, note that I removed track changes that show all our edits.

Thank you again for your time and efforts to review our manuscript and to provide insightful suggestions and edits.

Comment #1: The article concerns the stability of the Communication Function Classification System (CFCS) in a limited group of South Korea children with cerebral palsy (n=171). It repeats the only study present in literature about this topic made by Palisano et al. (2018) on a larger sample of 664 children. The authors cited Palisano et al. in the article and they got inspired to their methodology.

Response #1: Thank you for your understanding of the paper.

Comment #2: The results are similar but not completely overlapped an to my opinion the reasons of these different results are not well explained. Probably about communication there are also enviromental factors that must be keep in mind

Response #2: It was mentioned in the discussion (page 6, line 214~216).

Comment #3: Moreover, since that the South Korea sample is limited respect to that reported by Palisano et al., I suggested to modify the title specifying that it regards only that population.

Response #3: It was revised according to your comment (in title).

Comment #4: There are many mistakes about English that need a revision of the language (very often authors speak about children and use " was" instead of "were" in particular in the description of the tables).

Response #4: I appreciated your correction. It has been revised (in Results section).

Comment #5: There is a repetition at line 73 page 2 and a double verb at line 57 page 2 (are have).

Response #5: I appreciated your correction. It has been revised (page 2).

Comment #6: The authors must change the way to insert the reference at line 163 page 5 and line 34 page 1.

Response #6: I appreciated your correction. It has been revised (page 5, line 182).

Comment #7: I missed the sense of the phrase at line 186 page 6

Response #7: I revised manuscript more clearly (page 6, line 205 ~ 211).

Reviewer 2 Report

Brief summary

This research examined the stability ratings of  the Korean version of the Communication Function Classification System (CFCS) over 2 years with 171 children with cerebral palsy (CP) in South Korea. This is a unique contribution as only a few studies has examined stability in CFCS and this is the only study I am aware of with the Korean CFCS.

Broad comments

Strengths:

This is a large sample size for cerebral palsy research. The same occupational therapist classified the same child over the years. This has the important recommendation that the CFCS should be classified over the age span. The author is commended for studying an important clinical issue in cerebral palsy.

Concerns:

In the Introduction and/or Discussion sections, the author should discuss the role of augmentative and alternative communication in the CFCS classifications (see page 2 in the Korean CFCS) and whether stability is expected when an individual with CP becomes proficient in his AAC. For example, an individual with dysarthria who becomes proficient in AAC with switch scanning may rightly change classification from CFCS Level IV to more functional CFCS Level II. If faster access methods develop (e.g., brain-computer interfaces), the individual with dysarthria and CP may rightly demonstrate the most functional level of CFCS I.  This has most recently been discussed in the following journal articles: 
        Kristoffersson E, Dahlgren Sandberg A, Holck P. Communication ability and communication methods in children with cerebral palsy. Dev Med Child Neurol 2020.https://doi.org/10.1111/dmcn.14546
        Hidecker, MJC. The importance of communication classifications in cerebral palsy registers, Developmental Medicine & Child Neurology, 10.1111/dmcn.14571, 62, 8, (888-888), (2020).

Sometimes, the study is described as 2-years and sometimes as 3-years. If I understand correctly, this study collected data at beginning  (baseline), after one year, and after two year results which I interpret to be a 2-year study. Please edit throughout the paper to clarify the duration of the study and consistently describe this study throughout.

Analysis question: The age below 4 group needs further clarification. Was this group still below 4 at the final classification? If not, did that affect who changed levels or not by age? If yes, please clarify in the text.

Specific comments

Somewhere in the paper, perhaps in the references, the author should reference the Korean version of the CFCS and provide the web link for interested readers.

Introduction:

The first paragraph seems to be about the need for an assessment, then the second paragraph introduces the ICF, then the 3rd paragraph introduces the CFCS, GMFCS, and MACS. This may suggest to the reader that these classification are assessments, which they are not. For more details, please see Rosenbaum P, Eliasson AC, Hidecker MJ, Palisano RJ. Classification in childhood disability: focusing on function in the 21st century. J Child Neurol. 2014 Aug;29(8):1036-45. doi: 10.1177/0883073814533008. The author should consider either revising the first paragraph changing assessment to a term such as understand or in the third paragraph indicate that these classifications do not replace existing assessments but provide another tool to better understand a child’s functional communication performance.

Materials and Methods:

p. 2, line 80 Hidecker is female so please change “his” to “her”

p. 2, line 90 please indicate if the occupational therapist was blinded to previous CFCS ratings.

Results

p. 3, line 103 – p. 4, 110 avoid using the terms “higher” or “lower” with the CFCS levels due to the Roman numeral I indicating the most functional level and Roman numeral indicating the least functional communication level. Instead, note if the children’s communication was classified as more functional or less functional than the earlier rating.

Tables 1 through 3 readability could be improved if the diagonal numbers were bolded. This would easily show the reader the number of classifications that remained the same between time periods. To clarify the time periods between the ratings, could 1st be called “initial” or “baseline,” 2nd be called “Year 1” and 3rd be called “Year 2”?

Discussion:

Please discuss reasons that the children may increase functional levels such as the addition of AAC or improved speech and language due to intervention. Please discuss reasons that the children may decrease functional levels such as post-seizures or other co-morbidities or losing access to AAC. The author may want to refer to Hidecker MJC, Slaughter J, Abeysekara P, et al. Early Predictors and Correlates of Communication Function in Children With Cerebral Palsy. Journal of Child Neurology. 2018;33(4):275-285. doi:10.1177/0883073817754006

In limitations, could the OTs be misclassifying CFCS levels for children under 4 as discussed in  Cunningham, BJ, Rosenbaum, P, Hidecker MJC, Promoting consistent use of the CFCS Disability and Rehabilitation 2015 https://doi.org/10.3109/09638288.2015.1027009

Conclusion:

I like the recommendation that the CFCS level should be collected periodically. What does the author think about expanding the recommendation to include changes in communication methods, after communication interventions, as well as changes in co-morbidities such as seizures?

Author Response

In the revised manuscript, I have highlighted in blue where I have made changes; please, note that I removed track changes that show all our edits.

Thank you again for your time and efforts to review our manuscript and to provide insightful suggestions and edits.

Comment #1: This research examined the stability ratings of the Korean version of the Communication Function Classification System (CFCS) over 2 years with 171 children with cerebral palsy (CP) in South Korea. This is a unique contribution as only a few studies has examined stability in CFCS and this is the only study I am aware of with the Korean CFCS.

 Response #1: Thank you for your understanding of the paper.

Comment #2: This is a large sample size for cerebral palsy research. The same occupational therapist classified the same child over the years. This has the important recommendation that the CFCS should be classified over the age span. The author is commended for studying an important clinical issue in cerebral palsy.

Response #2: I am grateful for the evaluation of strength of this research.

Comment #3: In the Introduction and/or Discussion sections, the author should discuss the role of augmentative and alternative communication in the CFCS classifications (see page 2 in the Korean CFCS) and whether stability is expected when an individual with CP becomes proficient in his AAC. For example, an individual with dysarthria who becomes proficient in AAC with switch scanning may rightly change classification from CFCS Level IV to more functional CFCS Level II. If faster access methods develop (e.g., brain-computer interfaces), the individual with dysarthria and CP may rightly demonstrate the most functional level of CFCS I.  This has most recently been discussed in the following journal articles: 
        [50]Kristoffersson E, Dahlgren Sandberg A, Holck P. Communication ability and communication methods in children with cerebral palsy. Dev Med Child Neurol 2020.https://doi.org/10.1111/dmcn.14546
        Hidecker, MJC. The importance of communication classifications in cerebral palsy registers, Developmental Medicine & Child Neurology, 10.1111/dmcn.14571, 62, 8, (888-888), (2020).

Response #3: It was revised to your comments (page 7, line 220 ~ 225).

Comment #4: Sometimes, the study is described as 2-years and sometimes as 3-years. If I understand correctly, this study collected data at beginning (baseline), after one year, and after two year results which I interpret to be a 2-year study. Please edit throughout the paper to clarify the duration of the study and consistently describe this study throughout.

Response #4: I revised manuscript throughout according to your comments (page 3, line 121 ~ 122, throughout manuscript).

Comment #5: The age below 4 group needs further clarification. Was this group still below 4 at the final classification? If not, did that affect who changed levels or not by age? If yes, please clarify in the text.

Response #5: I agreed with your comment. More detail was inserted in the manuscript (page 3, line 120 ~121).

Comment #6: Somewhere in the paper, perhaps in the references, the author should reference the Korean version of the CFCS and provide the web link for interested readers.

Response #6: The link was inserted (page 2, line 86).

Comment #7: The first paragraph seems to be about the need for an assessment, then the second paragraph introduces the ICF, then the 3rd paragraph introduces the CFCS, GMFCS, and MACS. This may suggest to the reader that these classification are assessments, which they are not. For more details, please see Rosenbaum P, Eliasson AC, Hidecker MJ, Palisano RJ. Classification in childhood disability: focusing on function in the 21st century. J Child Neurol. 2014 Aug;29(8):1036-45. doi: 10.1177/0883073814533008. The author should consider either revising the first paragraph changing assessment to a term such as understand or in the third paragraph indicate that these classifications do not replace existing assessments but provide another tool to better understand a child’s functional communication performance.

Response #7: It was revised according to your comment (page 1, line 28 ~ 35).

Comment #8: p. 2, line 80 Hidecker is female so please change “his” to “her”

Response #8: I appreciated your correction. It has been revised page 2, line 87).

Comment #9: p. 2, line 90 please indicate if the occupational therapist was blinded to previous CFCS ratings.

Response #9: Thank you for your comment. It has been mentioned in the manuscript (page 3, line 106 ~ 107).

Comment #10: p. 3, line 103 – p. 4, 110 avoid using the terms “higher” or “lower” with the CFCS levels due to the Roman numeral I indicating the most functional level and Roman numeral indicating the least functional communication level. Instead, note if the children’s communication was classified as more functional or less functional than the earlier rating.

Response #10: I appreciated your correction. It has been revised (page 3, line 125 ~ page 5, line 156, throughout manuscript).

Comment #11: Tables 1 through 3 readability could be improved if the diagonal numbers were bolded. This would easily show the reader the number of classifications that remained the same between time periods. To clarify the time periods between the ratings, could 1st be called “initial” or “baseline,” 2nd be called “Year 1” and 3rd be called “Year 2”?

Response #11: The tables were revised according to comment. The terms for the measurement period were also revised according to comment throughout the document. I am grateful that the meaning has been conveyed more clearly (Table 1 ~ 3, throughout manuscript).

Comment #12: Please discuss reasons that the children may increase functional levels such as the addition of AAC or improved speech and language due to intervention. Please discuss reasons that the children may decrease functional levels such as post-seizures or other co-morbidities or losing access to AAC. The author may want to refer to Hidecker MJC, Slaughter J, Abeysekara P, et al. Early Predictors and Correlates of Communication Function in Children With Cerebral Palsy. Journal of Child Neurology. 2018;33(4):275-285. doi:10.1177/0883073817754006

Response #12: Thank you for kindly presenting references as well. The manuscript was revised according to the comments (page 7, line 222 ~ 225).

Comment #13: In limitations, could the OTs be misclassifying CFCS levels for children under 4 as discussed in Cunningham, BJ, Rosenbaum, P, Hidecker MJC, Promoting consistent use of the CFCS Disability and Rehabilitation 2015 https://doi.org/10.3109/09638288.2015.1027009

Response #13: It was mentioned according to your comment (page 7, line 237 ~238).

Comment #14: I like the recommendation that the CFCS level should be collected periodically. What does the author think about expanding the recommendation to include changes in communication methods, after communication interventions, as well as changes in co-morbidities such as seizures?

Response #14: Recommendation was extended according to your comment (page 7, line 252 ~ 254).

Reviewer 3 Report

Comments to the authors:

This is an interesting manuscript. The whole manuscript needs huge revisions for publication.

Following are major concerns:

  • Needs English writing consultation and spelling checking before submission. A lot of grammar errors, number errors and spelling errors in the whole text. For example, “a test-retest reliability of 0.82 % (CI = 0.74-0.90)” should be “a test-retest reliability of 0.82 (95%CI = 0.74-0.90) (P2, line 45), “compunction” should be “communication” (p6, line 180), “have an experience threating the children with CP more six months” should be “have an experience treating the children with CP more than six months” (P3, line 97), etc.
  • For most functional classification systems, such as GMFCS, MACS and CFCS, there are 5 age groups, <2、2~4、4~6、6~12 and 12~18 years. Still not clear why divide this study sample into two groups, < 4y and > 4ys. And the conclusion seems to come from Table 4, the Weighted Kappa = .620 between 1st and 3rd times. According to Discussion 2nd paragraph, lines 154-160, authors said “In general, the criteria for weighted kappa are a slight agreement between 0 and 0.20, a fair agreement between 0.21 and 0.40, a moderate agreement between 0.41 and 0.60, and a substantial agreement between 0.61 and 0.80, Interprets 0.81 to 1.00 as an almost perfect agreement [15].” Weighted Kappa = .620 belongs to “substantial agreement”.
  • Introduction: The first paragraph (P1, line 24-36) did not mention rationales and importance of this study. Why understanding stability of the CFCS for children with CP is important? What are the insufficient stability information of the CFCS in the previous existed studies? Definition of communication disorder could be found in the ICD 10. The statement of “One reason for this is the lack of a clear definition for communication disorders in children with CP” might be incorrect.    
  • Introduction, first paragraph, line 28-29, authors said “Therefore, there is a need for a comprehensive assessment that uses an informal evaluation based on direct observation or parent and teacher feedback “. I disagree that the FFCS is a comprehensive assessment that uses an informal evaluation. I would suggest CFCS is one of the multi-evaluation tools in natural setting. Functional classification is different from developmental tests or measurement; different measures has different purposes.
  • Materials and Methods. Suggest to add a paragraph about the content and existed reliability, stability, and validity information of the CFCS.
  • Materials and Methods. Suggest to add a paragraph to describe the demographic characteristics of assessors of the CFCS in three times. Do authors have pilot study about the inter-rater reliability or test-retest reliability of the CFCS of this study? If not, add this in the Discussion as one of the limitation of this study.
  • Materials and Methods. Suggest to add a paragraph to describe the statistical methods and its interpretation, such as agreement and weighted kappa. Suggest to use percent agreement instead of change ratio. In the Discussion, previous studies also use percent agreement.
  • Materials and Methods. Suggest to add percent agreement and weighted Kappa in Table 1 to 3. And describe briefly the characteristics of children with two-level difference between two times.
  • Should add limitation. Previous studies demonstrated the GMFCS level V of CP children is about 13-20%, this study sample showed that there were 50 % with level V (P2, 2nd paragraph of materials and methods). For severe ones, the stability of functional classification might be better.
  • Add ref (Himmelmann K, Lindh K, Hidecker MJ. Communication ability in cerebral palsy: A study from the CP register of western Sweden. Eur J Paediatr Neurol 2013;17(6):568-74.) In that article, showed that “Twenty-eight per cent were at the most functional CFCS level I, 13% at level II, 21% at level III, 10% at level IV and 28% at level V.”

Author Response

In the revised manuscript, I have highlighted in blue where I have made changes; please, note that I removed track changes that show all our edits.

Thank you again for your time and efforts to review our manuscript and to provide insightful suggestions and edits.

Comment #1: This is an interesting manuscript. The whole manuscript needs huge revisions for publication.

Response #1: I appreciate your review comments that can improve the quality of the paper, and I have accepted and revised all comments.

Comment #2: Needs English writing consultation and spelling checking before submission. A lot of grammar errors, number errors and spelling errors in the whole text. For example, “a test-retest reliability of 0.82 % (CI = 0.74-0.90)” should be “a test-retest reliability of 0.82 (95%CI = 0.74-0.90) (P2, line 45), “compunction” should be “communication” (p6, line 180), “have an experience threating the children with CP more six months” should be “have an experience treating the children with CP more than six months” (P3, line 97), etc.

Response #2: Thanks for your correction. I revised the manuscript and got English proofreading by native speaker (throughout manuscript).

Comment #3: For most functional classification systems, such as GMFCS, MACS and CFCS, there are 5 age groups, <2、2~4、4~6、6~12 and 12~18 years. Still not clear why divide this study sample into two groups, < 4y and > 4ys. And the conclusion seems to come from Table 4, the Weighted Kappa = .620 between 1st and 3rd times. According to Discussion 2nd paragraph, lines 154-160, authors said “In general, the criteria for weighted kappa are a slight agreement between 0 and 0.20, a fair agreement between 0.21 and 0.40, a moderate agreement between 0.41 and 0.60, and a substantial agreement between 0.61 and 0.80, Interprets 0.81 to 1.00 as an almost perfect agreement [15].” Weighted Kappa = .620 belongs to “substantial agreement”.

Response #3: Previous studies were referenced, and descriptions were added for this. Conclusion was revised according to your comments (page 3, line 20 ~ 121; page 6, line 179 ~ 181).

Comment #4: Introduction: The first paragraph (P1, line 24-36) did not mention rationales and importance of this study. Why understanding stability of the CFCS for children with CP is important? What are the insufficient stability information of the CFCS in the previous existed studies? Definition of communication disorder could be found in the ICD 10. The statement of “One reason for this is the lack of a clear definition for communication disorders in children with CP” might be incorrect.    

Response #5: I accepted our opinion and revised first paragraph (page 1, line 28 ~ 41).

Comment #5: Introduction, first paragraph, line 28-29, authors said “Therefore, there is a need for a comprehensive assessment that uses an informal evaluation based on direct observation or parent and teacher feedback “. I disagree that the FFCS is a comprehensive assessment that uses an informal evaluation. I would suggest CFCS is one of the multi-evaluation tools in natural setting. Functional classification is different from developmental tests or measurement; different measures has different purposes.

Response #5: I accepted our opinion and revised first paragraph (page 1, line 28 ~ 41).

Comment #6: Materials and Methods. Suggest to add a paragraph about the content and existed reliability, stability, and validity information of the CFCS.

Response #5: The content and existed reliability, stability, and validity information of the CFCS was inserted in materials and methods section (page 2, line 93 ~ 96).

Comment #7: Materials and Methods. Suggest to add a paragraph to describe the demographic characteristics of assessors of the CFCS in three times. Do authors have pilot study about the inter-rater reliability or test-retest reliability of the CFCS of this study? If not, add this in the Discussion as one of the limitation of this study.

Response #7: Unfortunately, no information was collected on demographic information of assessors of the CFCS. Limitations were given on the failure to provide demographic information along with pilot test about reliability (page 7, line 235 ~ 237).

Comment #8: Materials and Methods. Suggest to add a paragraph to describe the statistical methods and its interpretation, such as agreement and weighted kappa. Suggest to use percent agreement instead of change ratio. In the Discussion, previous studies also use percent agreement.

Response #8: I inserted describe the statistical methods and its interpretation. The results section was revised into percent agreement instead of change ratio (page 3, line 111 ~ 116; page 3, line 125 ~ page 5, line 156).

Comment #9: Materials and Methods. Suggest to add percent agreement and weighted Kappa in Table 1 to 3. And describe briefly the characteristics of children with two-level difference between two times.

Response #9: Tables were revised. The possible factors affected level change was discussed and the necessary to find out the specific variables was mentioned in terms of this study’s limitation (Table 1 ~ 3, page 6, line 216 ~ page 7, line 225).

Comment #10: Should add limitation. Previous studies demonstrated the GMFCS level V of CP children is about 13-20%, this study sample showed that there were 50 % with level V (P2, 2nd paragraph of materials and methods). For severe ones, the stability of functional classification might be better.

Response #12: I added the limitations according to your comment (page 7, line 237 ~ 240).

Comment #11: Add ref (Himmelmann K, Lindh K, Hidecker MJ. Communication ability in cerebral palsy: A study from the CP register of western Sweden. Eur J Paediatr Neurol 2013;17(6):568-74.) In that article, showed that “Twenty-eight per cent were at the most functional CFCS level I, 13% at level II, 21% at level III, 10% at level IV and 28% at level V.”

Response #11: I added that reference.

Reviewer 4 Report

-Small sample, especially for the under 4 year olds. 

-Could you potentially work with someone to help with clarity in the English language? Some quick edits for wording would be

Pg. 1

Line37 …(CFCS) instead of "which could" change to "is used to"

Line 42-43 take out "the developed"

Pg. 2 

For the study reported in lines 44-46, what are the ages of the study sample? 

Line 55 - the F is missing inn GMFCS

Line 57-58 - take out "are" before have not been tested 

line 70 ads an s on participants

71 change they to there

Line 80 - combine sentence starting with Hidecker with the one above by saying ...by Hidecker and his colleagues

When describing the therapists collecting the data, it would be good to know how many therapists were involved, and what % of kids were rated by the therapists at all 3 time intervals. 

Pg 3. I had to look into the table to understand that only 21 kids were below 4 and 150 were above. This could be clearly stated when describing the sample. 

When describing the tables, each time you have (XXX%) was rated.....Change the was to were. There are multiple example of this on page 3 and 4

Page 4. 

Table 2 - When describing the levels on the top of the table, would be more clear if it said third? This is the change between the 1st and third, right? 

Table 3 - When describing the Category on the table, After "age from 4 to 18", it would be more clear if it said second. this is the change between the 2nd and third, right? 

Page 5

The discussion was hard for me to follow, I think because of the language interpretation.  The first paragraph, all the sentences except the last one should be clarified. Unsure what you are getting at. 

Line 145 - I think you are saying with this sentence that the CFCS is similar for communication skills like GMFCS is for motor skills?? 

Line 153 add an s to "provide"

163 - There is an 11 (but not in []) i am thinking it should be [11] - also in this sentence, you should note if this data is in relation to the CFCS. 

I wonder if the discussion should mention that the change over the years in the children (especially the younger ones) could be a result of development of language. So some change is warranted (for those who improve). 

Author Response

In the revised manuscript, I have highlighted in blue where I have made changes; please, note that I removed track changes that show all our edits.

Thank you again for your time and efforts to review our manuscript and to provide insightful suggestions and edits.

Comment #1: Small sample, especially for the under 4 year olds. 

Response #1: Yes, it was. I mentioned about that in limitations (page 7, line 238 ~ 240).

Comment #2: Could you potentially work with someone to help with clarity in the English language? Some quick edits for wording would be

Response #2: Thanks for your correction. I revised the manuscript and got English proofreading by native speaker.

Comment #3: Pg. 1 Line37 …(CFCS) instead of "which could" change to "is used to"

Line 42-43 take out "the developed" Response #3: Thanks for your correction. They were corrected.

Comment #4: Pg. 2  For the study reported in lines 44-46, what are the ages of the study sample? 

Response #4: Age ranged was inserted (page 2, line 49).

Comment #5: Line 55 - the F is missing inn GMFCS

Response #5: Thanks for your correction. They were corrected.

Comment #6: Line 57-58 - take out "are" before have not been tested 

Response #6: Thanks for your correction. They were corrected.

Comment #7: line 70 ads an s on participants

Response #7: Thanks for your correction. They were corrected.

Comment #8: 71 change they to there

Response #8: Thanks for your correction. They were corrected.

Comment #9: Line 80 - combine sentence starting with Hidecker with the one above by saying ...by Hidecker and his colleagues

Response #9: It was revised according to your comment (page 2, line 86 ~ 87).

Comment #10: When describing the therapists collecting the data, it would be good to know how many therapists were involved, and what % of kids were rated by the therapists at all 3 time intervals. 

Response #10: It was revised according to your comment (page 3, line 99 ~ 100).

Comment #11: Pg 3. I had to look into the table to understand that only 21 kids were below 4 and 150 were above. This could be clearly stated when describing the sample. 

Response #11: It was stated (page 3, line 119 ~ 120).

Comment #12: When describing the tables, each time you have (XXX%) was rated.....Change the was to were. There are multiple example of this on page 3 and 4

Response #12: Thanks for your correction. They were corrected (page 6, line 216 ~ page 7, line 225).

Comment #13: Page 4.  Table 2 - When describing the levels on the top of the table, would be more clear if it said third? This is the change between the 1st and third, right? Table 3 - When describing the Category on the table, After "age from 4 to 18", it would be more clear if it said second. this is the change between the 2nd and third, right? 

Response #13: For more clarity, I am revising into the baseline, Year 1, and Year 2 according to the opinion of other reviewer (Table 1 ~ 3, throughout manuscript).

Comment #14: Page 5 The discussion was hard for me to follow, I think because of the language interpretation.  The first paragraph, all the sentences except the last one should be clarified. Unsure what you are getting at. Line 145 - I think you are saying with this sentence that the CFCS is similar for communication skills like GMFCS is for motor skills?? 

Response #14: First paragraph was revised (page 5, line 165 ~ 173).

Comment #15: Line 153 add an s to "provide"

Response #15: Thanks for your correction. They were corrected.

Comment #16: 163 - There is an 11 (but not in []) i am thinking it should be [11] - also in this sentence, you should note if this data is in relation to the CFCS. 

Response #16: Thanks for your correction. They were corrected.

Comment #17: I wonder if the discussion should mention that the change over the years in the children (especially the younger ones) could be a result of development of language. So some change is warranted (for those who improve). 

Response #17: The discussion was revised to reflect your comments (page 6, line 217 ~ page 7, line 220).

Round 2

Reviewer 3 Report

This is an interesting manuscript. The whole manuscript needs huge revisions for publication.

Following are major concerns:

  • Needs English writing consultation and data (values) checking before next submission. For example, Results, 3.1. Agreement ratio across three measurement points (P4, Lines 136-145)The statement of "The agreement ratio between year 1 and year 2 rating times The agreement ratio between year 1 and year 2 rating times according to age ranges 136 is shown in Table 2. " The Table 2 below the paragraph is " Table 2. The CFCS levels between baseline and year 2 rating. " An obviously error. Statement should be "The agreement rates between first and second rating among different age groups are shown in Table 2.” And the descriptions for Table 2 and Table 3 have to be revised.
  • Results, 3.1. Agreement ratio across three measurement points (P4, Line 128) “24 (14.0%) were rated in the less functional level”, “24” is incorrect, should change to “23”.
  • Introduction, third paragraph (P2, lines 57) “The MACS also confirmed the safety of 1,267 children with CP over five years” not understood.
  • Abstract (P1, line 7-8), “Interest in the prognosis of communication function has been a major concern for therapy 7 and intervention in children with cerebral palsy (CP).” And Conclusion (P7, line 256-257) “Understanding why children's CFCS levels change can help with their prognosis.” mentioned the importance of functional level assessment for prognosis. However, in the Introduction (P1, lines 24-35), authors did not mention the term of “prognosis”, only needs support, comprehensive assessment and “the degree of stability of the classification system can provide information on the possibility of child's function change” were described. The rationales and hypothesis of this study not described fully. Why it is important for this study, while there was one previous study of stability of CFCS (Palisano, R.J. et al., Dev Med Child Neurol 2018, 60, 1026-293 1032).  
  • Introduction (P2, line 69-70) “Based on a previous study on CFCS’s stability, here the CFCS stability was verified by evaluating data for children ≥ 4 years versus children < 4 years [11].” Should be described more detailed, and move to Materials and Methods section.
  • Materials and Methods (P2-3) need to be reorganized. Suggest to use subtitle as: participants, raters, data collection procedure, and statistical method. A lot of repetitive description in this section.
  • In Statistical method subsection of Materials and Methods section,
    • Suggest to say at beginning of this subsection as: ”Agreement rate and weighted Kappa were used in this study to examine the stability of the CFCS in three repetitive rating in two years”. Then describe in details about these two stability indexes.    
  • “Inter-rater agreement above 80% was of acceptable value [16].” (P3, lines 115-116) Is not relevant to this study. Is it agreement rate of stability?
    • Not consistent of description in P3 lines 112-115 with those in P2, line 68.
  • Results, “More functional level” change to “more severe functional level”, and“less functional level” change to “less severe functional level”. Not to confuse readers.
  • Results, please describe and discuss more about those children with 2-level difference after 1- or 2-year repetitive ratings.

Following are minor concerns:

  • Use terminology consistently, Agreement rate or ratio? Change rate or ratio?
  • Suggest to change as “Three rating: first rating, second rating and third rating. Each repetitive rating was with one-year interval.” Not use “baseline, year 1 and year 2” in Tables and text, confusing.
  • Suggest to change as “two subgroups: younger subgroup and older subgroup”. And the whole group.
  • The zero before a decimal point is known as a leading zero. If a value has the potential to exceed 1.0, use the leading zero. If a value can never exceed 1.0, do not use the leading zero. The values for weighted kappa or reliability coefficients should be without the leading zero.

Author Response

Thank you very much for giving us the opportunity to resubmit the manuscript. I am grateful to you, and the reviewers for insightful suggestions and comments. I have accepted or addressed all the suggested edits and comments. I hope that we have adequately strengthened out manuscript.

I describe how I addressed each suggested edit and comment. In the revised manuscript, I have highlighted in blue where we have made changes; please, note that we removed track changes that show all our edits.

Thank you again for your time and efforts to review our manuscript and to provide insightful suggestions and edits.

Comment #1: Needs English writing consultation and data (values) checking before next submission. For example, Results, 3.1. Agreement ratio across three measurement points (P4, Lines 136-145). The statement of "The agreement ratio between year 1 and year 2 rating times The agreement ratio between year 1 and year 2 rating times according to age ranges 136 is shown in Table 2. " The Table 2 below the paragraph is " Table 2. The CFCS levels between baseline and year 2 rating. " An obviously error. Statement should be "The agreement rates between first and second rating among different age groups are shown in Table 2.” And the descriptions for Table 2 and Table 3 have to be revised.

Response #1: The descriptions were revised for Table 2 and Table 3 as followed.

The agreement rates between the first and third ratings among different age groups are shown in Table 2. In the whole group, the agreement rate between the first rating and ratings was 66.1% (n = 113). In total, 21 children with CP (12.3%) were rated in the more severe functional level and 37 (21.6%) were rated in the less severe functional level during the third rating than in the first rating. Regarding the younger group, the change rate was 42.9% (n = 9). Four children with CP (19.0%) were rated in the more severe functional level and five children with CP (23.9%) were rated in the less severe functional level during the third rating than in the first rating. Regarding the older group, the change rate was 28.7% (n = 49). Totally, 26 children with CP (17.4%) were rated in the more functional severe level and 23 (15.3%) were rated in the less severe functional level.    

Comment #2:  Results, 3.1. Agreement ratio across three measurement points (P4, Line 128) “24 (14.0%) were rated in the less functional level”, “24” is incorrect, should change to “23”.

Response #2: It was corrected.

In total, 30 children with CP (17.6%) were rated in the more severe functional level and 23 (13.5%) were rated in the less severe functional level during the second rating than in the first rating.

Comment #3: Introduction, third paragraph (P2, lines 57) “The MACS also confirmed the safety of 1,267 children with CP over five years” not understood.

Response #3: Thank you for your check. It was changed into stability.

Comment #4: Abstract (P1, line 7-8), “Interest in the prognosis of communication function has been a major concern for therapy 7 and intervention in children with cerebral palsy (CP).” And Conclusion (P7, line 256-257) “Understanding why children's CFCS levels change can help with their prognosis.” mentioned the importance of functional level assessment for prognosis. However, in the Introduction (P1, lines 24-35), authors did not mention the term of “prognosis”, only needs support, comprehensive assessment and “the degree of stability of the classification system can provide information on the possibility of child's function change” were described. The rationales and hypothesis of this study not described fully. Why it is important for this study, while there was one previous study of stability of CFCS (Palisano, R.J. et al., Dev Med Child Neurol 2018, 60, 1026-293 1032).  

Response #4: Abstract and introduction for rationale was revised.

Abstract: Interest in the prognosis of skill level has been an important issue among children with cerebral palsy (CP).

Introduction: Whether or not the child's skill level will change because of a change in prognosis, decision making and counseling with parents is an important issue, and the degree of stability of the classification system can provide information regarding the possibility of changes in functioning in the child. The stability of the classification system indicates whether children with CP maintain the same level of functioning over time or whether they can be reclassified to different levels over time [14].

Comment #5: Introduction (P2, line 69-70) “Based on a previous study on CFCS’s stability, here the CFCS stability was verified by evaluating data for children ≥ 4 years versus children < 4 years [11].” Should be described more detailed, and move to Materials and Methods section.

Response #5: That sentence was moved into materials and methods section and revised.

Comment #6: Materials and Methods (P2-3) need to be reorganized. Suggest to use subtitle as: participants, raters, data collection procedure, and statistical method. A lot of repetitive description in this section.

Response #6: Materials and Methods section was reorganized according to your comments.

  1. Materials and Methods

2.1. Participants

Totally, 171 children with CP (mean = 10.9 years, SD = 4.6 years) participated in this study. Participants attended a convalescent or rehabilitation center for disabled individuals or a special school for physical disabilities in South Korea. There were 99 boys (57.9%) and 72 girls (42.1%). The age range was 2 to 18 years. Totally, 21 children with CP were below 4 years and 150 were older. The parents of all children agreed to participate in this study. The types of CP in the children were spastic (81.0%), dyskinetic/athetotic (6.8%), ataxic (3.4%), and hypotonic (8.8%). The participants were classified using the GMFCS: 19 (11.1%) were classified into Level I, 24 (14.0%) into Level II, 18 (10.5%) into Level III, 24 (14.0%) into Level IV, and 86 (50.3%) into Level V. The present study was approved by the Research Ethics Board of Jeonju University (Jeonju University IRB-1041042-2013-1).

2.2. Measurement

To classify their communication function, the CFCS was used. CFCS is a tool developed with support from the National Institute of Health (NIH), the Cerebral Palsy International Research Foundation, and The Hearst Foundation. We used the Korean version of the CFCS (http://cfcs.us/wp-content/uploads/2018/11/CFCS_Korean.pdf) [7] developed by Hidecker and her colleagues [4]. The CFCS content was as follows: Level 1 child with CP is an effective sender and receiver with unfamiliar and familiar partners. Level 2 child with CP is an effective but slower paced sender and/or receiver with unfamiliar and familiar partners. Level 3 child with CP is an effective sender and receiver with familiar partners. Level 4 child with CP is sometimes an effective sender and receiver with familiar partners. Level 5 child with CP is seldom an effective sender and receiver even with familiar partners. The inter-rater reliability of CFCS has been reported from .66 to .98 [4, 16]. Hidecker et al. [17] reported the validity of the CFCS in preschool children with speech and language disorders. The stability of the CFCS was reported by previous studies. Palisano et al. [13] reported the kappa coefficients to be .57-.77 in 664 children with CP.

2.3. Raters

Data regarding the CFCS were collected by 18 occupational therapists who had treated children with CP for more than 6 months. One occupational therapist evaluated 3-9% of the children with CP. Although occupational therapists met the eligibility requirements for administering the CFCS evaluations, all therapists were trained specifically for using these evaluation tools. They were encouraged to seek clarification from the investigators regarding any question or problem arising during the assessments. The amount of time it took to complete the CFCS was less than 5 minutes because the occupational therapists were familiar with children with CP. The same evaluators performed the assessments across the three measurements and were blinded to the previous CFCS ratings.

2.4. Data collection procedure

The data were collected once a year over two years during the same month. For example, if data were collected on September 1 of the first year, data were collected between September 1 and September 31 the next year. After the third measurement of evaluation, all collected data were subjected to central statistical monitoring.

2.5. Statistical method

Agreement rate of the CFCS and weighted kappa were used in this study to examine the stability of the CFCS in three repetitive ratings in two years. Weighted kappa was performed to investigate the change of the CFCS across three times measurements. In general, the criteria for the weighted kappa are: a slight agreement between 0 and .20, a fair agreement between .21 and .40, a moderate agreement between .41 and .60, a substantial agreement between .61 and .80, and .81 to 1.00 interprets as an almost perfect agreement [18]. Agreement rate of stability above 80% was of acceptable value [19].

Based on a previous study on CFCS’s stability, here the CFCS stability was verified by evaluating data for children ≥ 4 years versus children < 4 years [13]. The analysis was completed for about 171 children (whole group) with 21 being below 4 years (younger group) and 150 above 4 years (older group). The age criterion for dividing two subgroups was set to 4 years with reference to the first study to find out the stability of the CFCS [13]. The age reference point for dividing the groups was 4 years at first rating.

Comment #7: In Statistical method subsection of Materials and Methods section, Suggest to say at beginning of this subsection as: ”Agreement rate and weighted Kappa were used in this study to examine the stability of the CFCS in three repetitive rating in two years”. Then describe in details about these two stability indexes.    

Response #7: That sentence was inserted.

In general, the criteria for the weighted kappa are: a slight agreement between 0 and .20, a fair agreement between .21 and .40, a moderate agreement between .41 and .60, a substantial agreement between .61 and .80, and .81 to 1.00 interprets as an almost perfect agreement [18]. Agreement rate of stability above 80% was of acceptable value [19].

Comment #8: “Inter-rater agreement above 80% was of acceptable value [16].” (P3, lines 115-116) Is not relevant to this study. Is it agreement rate of stability?

Response #8: Yes, it is. It was revised.

Agreement rate of stability above 80% was of acceptable value [19].

Comment #9: Not consistent of description in P3 lines 112-115 with those in P2, line 68.

Response #9: I was sorry for confusing statement. Sentence in introduction was deleted.

Comment #10: Results, “More functional level” change to “more severe functional level”, and“less functional level” change to “less severe functional level”. Not to confuse readers.

Response #10: It was revised according to your comments.

In total, 30 children with CP (17.6%) were rated in the more severe functional level and 23 (13.5%) were rated in the less severe functional level during the second rating than in the first rating. Regarding the younger group, the change rate was 33.3% (n = 7). Three children with CP (14.3%) were rated in the more severe functional level and four children with CP (19.0%) were rated in the less severe functional level during the second rating than in the first rating. Regarding the older group, the change rate was 30.7% (n = 46). Totally, 27 children with CP (18.0%) were rated in the more severe functional level and 19 (12.7%) were rated in the less severe functional level.

Comment #11: Results, please describe and discuss more about those children with 2-level difference after 1- or 2-year repetitive ratings.

Response #12: It was described in discussion.

In this study, the number of children with CP that moved into a 2-level difference was seven both between the first and second rating and also between the first and third rating. The number of children with a 2-level difference between the second and third rating was five. The case that moved into a less severe functional level was higher than those that moved into a more severe functional level. As a factor influencing the change to a less severe functional level, the effect of child maturity and intervention is expected. It is necessary to also consider the development of communication skills through maturity as a cause that may affect the change in the level of CFCS in children. If the child received AAC-based interventions or other effective speech and language interventions, this would have affected the less severe functional changes in CFCS levels. As a factor influencing the change to a more severe functional level, post-seizures, other co-morbidities and losing access to AAC could be a possibility.

Comment #12: Following are minor concerns: Use terminology consistently, Agreement rate or ratio? Change rate or ratio?

Response #11: It was revised according to reviewer’s comments throughout manuscript. Example was as followed.

The agreement rate between the second and third ratings according to age ranges is shown in Table 3. In the whole group, the agreement rate between the second and third ratings was 80.1% (n = 137). In total, 13 children with CP (7.6%) were rated in the more severe functional level and 21 (13.3%) were rated in the less severe functional level during the third rating than in the second rating. Regarding the younger group, the change rate was 33.3% (n = 7). Two children with CP (9.5%) were rated in the more severe functional level and five children with CP (23.8%) were rated in the less severe functional level during the third rating than in the second rating. Regarding the older group, the change rate was 18.0% (n = 27). Totally, 11 children with CP (7.3%) were rated in the more severe functional level and 26 (10.7%) were rated in the less severe functional level.    

Comment #13: Suggest to change as “Three rating: first rating, second rating and third rating. Each repetitive rating was with one-year interval.” Not use “baseline, year 1 and year 2” in Tables and text, confusing.

Response #13: It was revised according to reviewer’s comments throughout manuscript. Example was as followed.

The agreement rate between the second and third ratings according to age ranges is shown in Table 3. In the whole group, the agreement rate between the second and third ratings was 80.1% (n = 137). In total, 13 children with CP (7.6%) were rated in the more severe functional level and 21 (13.3%) were rated in the less severe functional level during the third rating than in the second rating. Regarding the younger group, the change rate was 33.3% (n = 7). Two children with CP (9.5%) were rated in the more severe functional level and five children with CP (23.8%) were rated in the less severe functional level during the third rating than in the second rating. Regarding the older group, the change rate was 18.0% (n = 27). Totally, 11 children with CP (7.3%) were rated in the more severe functional level and 26 (10.7%) were rated in the less severe functional level.    

Comment #14: Suggest to change as “two subgroups: younger subgroup and older subgroup”. And the whole group.

Response #14: It was revised according to reviewer’s comments throughout manuscript. Example was as followed.

The agreement rate between the second and third ratings according to age ranges is shown in Table 3. In the whole group, the agreement rate between the second and third ratings was 80.1% (n = 137). In total, 13 children with CP (7.6%) were rated in the more severe functional level and 21 (13.3%) were rated in the less severe functional level during the third rating than in the second rating. Regarding the younger group, the change rate was 33.3% (n = 7). Two children with CP (9.5%) were rated in the more severe functional level and five children with CP (23.8%) were rated in the less severe functional level during the third rating than in the second rating. Regarding the older group, the change rate was 18.0% (n = 27). Totally, 11 children with CP (7.3%) were rated in the more severe functional level and 26 (10.7%) were rated in the less severe functional level.    

Comment #15: The zero before a decimal point is known as a leading zero. If a value has the potential to exceed 1.0, use the leading zero. If a value can never exceed 1.0, do not use the leading zero. The values for weighted kappa or reliability coefficients should be without the leading zero.

Response #15: It was revised according to reviewer’s comments throughout manuscript. Example was as followed.

In terms of the CFCS levels’ stability, the weighted kappa coefficients were .757 to .873 in the whole group (Table 1-3).

Reviewer 4 Report

Really nice job with the revision. Only a couple minor suggestions: 

pg. 1, line 44 change the word "evaluating" to classifying after ...trend in

pg. 2, line 45 change "it was" to "they were"

Pg. 2, line 49 add "on the CFCS" after ....2 to 18 years on the CFCS

Pg. 2, line 83 change "evaluate" to "classify"

Author Response

Thank you very much for giving us the opportunity to resubmit the manuscript. I am grateful to you, and the reviewers for insightful suggestions and comments. I have accepted or addressed all the suggested edits and comments. I hope that we have adequately strengthened out manuscript.

I describe how I addressed each suggested edit and comment. In the revised manuscript, I have highlighted in blue where we have made changes; please, note that we removed track changes that show all our edits.

Thank you again for your time and efforts to review our manuscript and to provide insightful suggestions and edits.

Comment #1: pg. 1, line 44 change the word "evaluating" to classifying after ...trend in

Response #1: “evaluating” was change into “classifying”.

Comment #2: pg. 2, line 45 change "it was" to "they were"

Response #2: It was changed into “they were”.

Comment #3: Pg. 2, line 49 add "on the CFCS" after ....2 to 18 years on the CFCS

Response #3: “on the CFCS” was inserted.

Comment #4: Pg. 2, line 83 change "evaluate" to "classify"

Response #4: “evaluate” was changed into “classify”